# Effects of Sodium Nitroprusside on Lipopolysaccharide-Induced Inflammation and Disruption of Blood–Brain Barrier

**DOI:** 10.3390/cells13100843

**Published:** 2024-05-15

**Authors:** Nuria Seoane, Aitor Picos, Sandra Moraña-Fernández, Martina Schmidt, Amalia Dolga, Manuel Campos-Toimil, Dolores Viña

**Affiliations:** 1Physiology and Pharmacology of Chronic Diseases (FIFAEC) Center for Research in Molecular Medicine and Chronic Diseases (CiMUS), University of Santiago de Compostela, 15782 Santiago de Compostela, Spain; nuria.seoane@rai.usc.es (N.S.); aitor.picos@rai.usc.es (A.P.); sandra.morana.fernandez@usc.es (S.M.-F.); mdolores.vina@usc.es (D.V.); 2Department of Molecular Pharmacology, University of Groningen, 9713 AV Groningen, The Netherlands; m.schmidt@rug.nl (M.S.); a.m.dolga@rug.nl (A.D.); 3Department of Pharmacology, Pharmacy and Pharmaceutical Technology, University of Santiago de Compostela, 15782 Santiago de Compostela, Spain

**Keywords:** bEnd.3 cells, blood–brain barrier, inflammation, lipopolysaccharide, neurodegenerative diseases, sodium nitroprusside

## Abstract

In various neurodegenerative conditions, inflammation plays a significant role in disrupting the blood–brain barrier (BBB), contributing to disease progression. Nitric oxide (NO) emerges as a central regulator of vascular function, with a dual role in inflammation, acting as both a pro- and anti-inflammatory molecule. This study investigates the effects of the NO donor sodium nitroprusside (SNP) in protecting the BBB from lipopolysaccharide (LPS)-induced inflammation, using bEnd.3 endothelial cells as a model system. Additionally, Raw 264.7 macrophages were employed to assess the effects of LPS and SNP on their adhesion to a bEnd.3 cell monolayer. Our results show that LPS treatment induces oxidative stress, activates the JAK2/STAT3 pathway, and increases pro-inflammatory markers. SNP administration effectively mitigates ROS production and IL-6 expression, suggesting a potential anti-inflammatory role. However, SNP did not significantly alter the adhesion of Raw 264.7 cells to bEnd.3 cells induced by LPS, probably because it did not have any effect on ICAM-1 expression, although it reduced VCAM expression. Moreover, SNP did not prevent BBB disruption. This research provides new insights into the role of NO in BBB disruption induced by inflammation.

## 1. Introduction

The blood–brain barrier (BBB) is a highly selective interface between the blood and the brain, maintaining a controlled microenvironment for neural signaling in the central nervous system (CNS). The neurovascular unit (NVU), consisting of endothelial cells, astrocytes, pericytes, microglia, neurons, and extracellular matrix, ensures the proper functioning of the BBB through the precise interactions among its components. Thus, only lipophilic molecules and a few hydrogen bond donors and/or acceptors can diffuse across it under physiological conditions [1].

Emerging evidence supports the critical role of BBB dysfunction in the pathogenesis of various neurodegenerative disorders, such as Alzheimer’s disease, Parkinson’s disease, amyotrophic lateral sclerosis, multiple sclerosis, and Huntington’s disease [2]. However, it is challenging to determine whether BBB dysfunction can be a causal factor or a consequence of the disorder, arising at some point during the development of the disease. In any case, BBB dysfunction has been identified as an early biomarker in neurodegenerative diseases, supported by strong evidence from neuroimaging, postmortem studies, and cerebrospinal fluid analysis [3].

Among the situations that can alter the functionality of the BBB and pathologically increase its permeability is inflammation [4]. A significant percentage of the population in developed countries experiences mild systemic inflammation linked to metabolic syndrome, insulin resistance, type 2 diabetes, high blood pressure, dyslipidemia, and obesity. In fact, these factors have been recognized as risk factors for various neurodegenerative conditions [5,6].

Upon disruption of the BBB’s integrity, cytokines and immune cells infiltrate the CNS, triggering glial cell activation and inducing alterations in the extracellular environment. These changes lead to secondary inflammation and additional BBB impairment, resulting in the leakage of plasma proteins and neurotoxic substances. Although CNS inflammation can stem from BBB dysfunction, an inflammatory response within the brain can also contribute to endothelial cell damage and increased BBB permeability [7].

Numerous findings point to at least two primary neuroinflammatory signaling cascades, the prostaglandin and/or nitric oxide (NO) pathways, which could mediate aspects of blast-induced delayed BBB disruption. However, the specific role of NO in this context remains relatively understudied, and we will primarily focus on investigating this less-explored aspect of BBB regulation.

A basal production of NO is required for physiological cell regulation, whereas an excess of NO can be cytotoxic to the organism. Endothelial NOS (eNOS) is a calcium/calmodulin-dependent isoform that releases a small amount of NO to control the local blood flow through soluble guanylyl cyclase (sGC) activation in vascular smooth muscle cells. Physiologic NO levels reduce oxidative stress by inhibiting anion superoxide production via inactivation of NADH/NADPH oxidase and increase endogenous antioxidant ability by inducing endothelial superoxide dismutase (SOD) [8].

Unlike eNOS, the inducible isoform of NOS (iNOS), a calcium-independent isoform, is stimulated and produced by inflammatory responses, generating a large quantity of NO, which is strongly pro-oxidant, disrupting cell membranes, impairing cell signaling, and reducing cell survival [9]. Inappropriate release of NO has been reported as a mediator of hypoxia-induced BBB breakdown, and both nonselective NOS inhibitor (L-NAME) and selective iNOS inhibitor (1400 W) attenuated the deleterious effects of hypoxia on paracellular permeability in brain microvessel endothelial cells [10].

Treatment of endothelial cells with NO donors under inflammatory conditions has yielded controversial results. It has been described that NO donors have an anti-inflammatory role in human vascular endothelial cells because they inhibit cytokine-induced NF-κB activation and adhesion molecule expression [11]. Additionally, sodium nitroprusside (SNP), a NO donor, prevents the detrimental effects of glucose on the neurovascular unit and behavior in zebrafish [12]. Recently, it has also been reported that S-nitroso-N-acetylpenicillamine (SNAP), an external source of NO, induces a dose-dependent increase in transendothelial electrical resistance (TEER) values in human brain microvascular endothelial cells (HBMECs). However, the mRNA expression of claudin-5 and claudin-1 in SNAP-treated HBMECs remains unchanged. Likewise, the protein levels of claudin-5 and VE-Cadherin were not altered [13]. 

On the contrary, in different cultures of brain microvascular endothelial cells, it has been demonstrated that a NO donor, such as SNP, increases blood–brain barrier (BBB) permeability by disrupting ZO-1 [14]. ZO-1 is a cytoplasmic scaffold protein normally localized at the cell membrane under physiological conditions that connects tight junction (TJ) membrane proteins to actin filaments, thereby facilitating their binding to the actin cytoskeleton [15]. Also, an increase in BBB permeability has been observed in vivo following treatment with NO donors [16].

Herein, the main objective of this study is to investigate the potential protective effects of SNP on LPS-challenged bEnd.3 cell cultures, used as a BBB model, in order to evaluate its impact on permeability and disruption under inflammatory conditions.

## 2. Materials and Methods

### 2.1. Reagents

Mouse microvascular cerebral endothelial cells bEnd.3 and murine macrophage cell line Raw 264.7 were purchased from ATCC (Manassas, VA, USA). Dulbecco’s Modified Eagle Medium (DMEM) with a low glucose concentration (5.5 mM) was obtained from Biowest (Nuaillé, France). Fetal bovine serum (FBS), penicillin/streptomycin, and 0.05% trypsin-EDTA were obtained from Gibco (ThermoFisher Scientific, Waltham, MA, USA). Sodium nitroprusside (SNP), lipopolysaccharide (LPS), 3-(4,5-dimethyltiazol-2-yl)-2,5-diphenyltetrazolium bromide (MTT), sulfanilamide, N-(1-naphthyl)ethylenediamine (NED), and Triton-X-100 were obtained from Sigma-Aldrich (St. Quentin Fallavier, France). Dimethylsulfoxide (DMSO) was obtained from Fisher Scientific (USA). CM-H2DCFDA, TRIzol reagent, 5× First Strand Buffer, MgCl_2_, random primers, RNaseOUT ribonuclease inhibitor, M-MLV reverse transcriptase, anti-claudin-5 mouse monoclonal antibody (Cat # 35-2500), anti-ZO-1 monoclonal antibody (Cat # 33-9100), Calcein AM, DyLight^TM^ 488-conjugated anti-mouse antibody (Cat # 35502), and Alexa Fluor 680-conjugated anti-mouse antibody (Cat # A21058) were obtained from Invitrogen (ThermoFisher Scientific, Waltham, MA, USA). dATP, dTTP, dGTP, and dCTP were obtained from Promega (Promega Biotech Ibérica, Madrid, Spain). The Luminaris Color HiGreen qPCR Master Mix kit was obtained from Thermo Scientific (ThermoFisher Scientific, Waltham, MA, USA). Phenylmethylsulfonyl fluoride (PMSF), sodium orthovanadate, sodium fluoride and aprotinin were obtained from MP Biomedicals. Anti-β-actin mouse monoclonal antibody (Cat # sc-47778) was obtained from Santa Cruz Biotechnology (Santa Cruz, CA, USA). Anti-phospho-STAT3 rabbit (Cat # 9145), HRP-conjugated anti-mouse (Cat # 7076S) and HRP-conjugated anti-rabbit monoclonal (Cat # 7074S) antibodies were obtained from Cell Signaling (Waltham, MA, USA). Anti-STAT3 mouse monoclonal antibody (Cat # ab50761) and phalloidin-iFluor 594 reagent were obtained from Abcam, Cambridge, UK. Anti-iNOS mouse monoclonal antibody (Cat # 610432) was obtained from BD (Milpitas, NJ, USA). Additionally, 18-well μ-slides were obtained from Ibidi (Gräfelfing, Germany). 

### 2.2. Cell Culture and Treatment

bEnd.3 cells and Raw 264.7 cells were grown in DMEM containing 10% (*v*/*v*) FBS and 1% (*v*/*v*) penicillin/streptomycin at 37 °C in a humidified atmosphere containing 5% CO_2_. For all the experiments, confluent bEnd.3 cells were subjected to inflammation induced by 24 h LPS treatment (10 µg/mL). To assess the effect that increasing NO levels has on cellular function and barrier integrity, SNP (100 µM) was used.

### 2.3. Cell Viability Assay

Cell viability was assessed using the MTT assay. Briefly, bEnd.3 cells were seeded into 96-well plates at a density of 15,000 cells/well. Once confluent, the cultures were exposed to inflammatory stress as well as to SNP (100 µM) treatment for 24 h. Afterwards, 10 µL of MTT solution (5 mg/mL) were added, and the cells were incubated for 2 h at 37 °C. The medium was replaced with 100 µL of DMSO to dissolve the formazan crystals by gently tapping. Absorbance at 570 nm was detected using a FLUOstar OPTIMA microplate reader (BMG LabTech, Ortenberg, Germany). The results were expressed as percentages of the respective controls.

### 2.4. xCELLigence Measurements

Alterations in cellular morphology and proliferation were assessed using the xCELLigence ^®^RTCA MP System (ACEA BIO, San Diego, CA, USA), which is a label-free cell-based assay suitable for continuous monitoring of biological processes in living cells [17]. bEnd.3 cells were seeded into 96-well E-plates at a density of 8000 cells/well. After 24 h, bEnd.3 cells were exposed to inflammatory stress stimuli as well as to SNP treatment. Cellular impedance was assessed every 30 min for 24 h and represented as normalized Cell Index (nCI), which was defined before the application of the different experimental conditions (control, LPS, SNP, and co-treatment of LPS and SNP) as the starting point (t: 0 h) of the experiment.

### 2.5. NO Production

To determine NO production by bEnd.3 cells, the Griess assay was performed. Confluent cultures of bEnd.3 cells in 96-well plates were exposed to LPS in the presence or absence of SNP. After treatment completion, 25 µL of 1% (P/V) sulfanilamide in 5% (*v*/*v*) H_3_PO_4_ were added to 50 µL of supernatant, and 10 min later, 25 µL of 0.1% (P/V) NED were added. After 10 min incubation at room temperature, absorbance was detected at 540 nm using a microplate reader.

### 2.6. Intracellular Reactive Oxygen Species (ROS) Measurement

After exposure to the different treatments, cells were incubated with a serum-free medium containing CM-H2DCFDA (5 µM) for 30 min at 37 °C with 5% CO_2_. After removing the CM-H2DCFDA solution, wells were replenished with normal growth medium, and cell cultures were maintained at 37 °C with 5% CO_2_ for 30 min. Cells were harvested by trypsinization, and fluorescence (λex: 485 nm, λem: 520 nm) was measured using a flow cytometer via the CytoFLEXS benchtop flow cytometer (Beckman Coulter Life Sciences, Brea, CA, USA).

### 2.7. Quantitative Reverse Transcription Polymerase Chain Reaction (RT-qPCR)

Total RNA was extracted from bEnd.3 cells after 24 h of incubation with LPS in the presence or absence of SNP using TRIzol reagent. RNA was reversely transcripted into cDNA using a mix of MgCl_2_ (2.5 mM), dNTPs (0.5 mM), random primers (17 ng/μL), RNase OUT ribonuclease inhibitor (33.3 U/mL), and M-MLV reverse transcriptase (13.3 U/mL) in 5× First Strand Buffer. qPCR was performed using the Luminaris Color HiGreen qPCR Master Mix Kit (ThermoFisher Scientific, Waltham, MA, USA) according to the manufacturer’s instructions. The relative expression of interleukin-6 (IL-6), vascular cell adhesion molecule 1 (VCAM-1), and intracellular cell adhesion molecule 1 (ICAM-1) mRNA was normalized to the expression of β-actin. The primers for IL-6, VCAM-1, ICAM-1, and β-actin were listed as follows: Mouse IL-6: (F) GGGACTGATGCTGGTGACAA, (R) AGCATTGGAAATTGGGGTAGGA; Mouse VCAM-1: (F) CCC AAG GAT CCA GAG ATT CA, (R) TAA GGT GAG GGT GGC ATT TC; Mouse ICAM-1: (F) GAA GGT GGT TCT TCT GAG CG, (R) GTC TGC TGA GAC CCC TCT TG; Mouse β-actin: (F) CTGAGAGGGAAATCGTGCGT, (R) AGGGTGTAAAACGCAGCTCAG.

### 2.8. Transendothelial Electrical Resistance Measurement (TEER)

bEnd.3 cells were seeded in the apical compartment of culture inserts for 24-well plates with 0.4 µm pores at a density of 10^5^ cells/cm^2^. The formation of a tight monolayer was evaluated by measuring TEER values between days 3 and 7 of culture using a Millicell ERS-2 Voltohmmeter (MilliporeSigma, Burlington, MA, USA). TEER values were expressed in Ω·cm^2^ after subtracting the TEER value of the blank insert and multiplying by the membrane area. Once the values were stable for 24 h, the medium was changed to 3% FBS DMEM, and monolayers were exposed to LPS and SNP for 24 h, after which TEER measurement was performed one last time. 

### 2.9. Western Blot Analysis

After the exposure of cells with or without SNP treatment to LPS, cells were lysed in lysis buffer [Tris-HCl pH 7.5 (50 mM), NaCl (150 mM), EDTA (1 mM), 1% (*v*/*v*) Triton X-100 in miliQ water], supplemented with a cocktail of protease inhibitors (PMSF, sodium orthovanadate, sodium fluoride, and aprotinin), and centrifuged for 15 min at 4 °C at 7260× *g*. Lysates were denatured at 95 °C for 6 min in sample buffer, and proteins were separated on an 8–15% SDS-PAGE gel by electrophoresis. Proteins were then transferred onto a nitrocellulose membrane and blocked with 5% BSA in TBS-Tween 0.2% for 2 h. Membranes were incubated with primary antibodies: mouse anti-ZO-1 (1:1000), mouse anti-iNOS (1:1000), mouse anti-claudin-5 (1:2000), mouse anti-STAT-3 (1:1000), rabbit anti-p-STAT-3 (1:1000), and mouse anti-β-actin (1:1000) overnight at 4 °C. For claudin-5, membranes were washed with TBS-Tween 0.2% for 5 min 4–6 times and incubated with secondary antibody (DyLight 488TM anti-mouse) for 1 h and then washed again prior to detection of immunoreactive bands using ChemiDoc (Bio-Rad, Hercules, CA, USA). For the rest of the antibodies, washing with TBS-Tween 0.2% was followed by incubation with horseradish peroxidase (HRP)-conjugated goat anti-rabbit IgG or goat anti-mouse IgG. The immunoreactive bands were visualized using PierceTM ECL Western Blotting Substance.

### 2.10. Immunocytochemistry

bEnd.3 cells were seeded on 18-well μ-slides. After confluence was reached, the cell monolayer was exposed to LPS (10 μg/mL) for 24 h in the presence or absence of SNP (100 μM). After treatment completion, bEnd.3 cells were fixed in 4% PFA for 15 min, permeabilized with 0.1% Triton-X-100 in PBS for 5 min, and blocked in 1% BSA in PBS for 1 h at room temperature. Slides were incubated with the primary antibody against ZO-1 (1:100) and Claudin-5 (1:100) overnight at 4 °C, followed by incubation with anti-mouse Alexa Fluor 680-conjugated secondary antibody (1:1000), phalloidin-iFluor 594 reagent (1:1000), and Hoechst (1:5000) for 1 h at room temperature. Cells were sealed with mounting medium and observed under a TCS SP5 X confocal microscope (Leica Biosystems, Wetzlar, Germany).

### 2.11. Cell Adhesion Assay

The cell adhesion assay was conducted following the method described by Hu et al. [18], with slight modifications. Confluent cultures of bEnd.3 cells in 12-well plates were treated with LPS (10 µg/mL) and/or SNP (100 µM) for 24 h. Raw 264.7 macrophages were stained with Calcein AM (2 µM) at 37 °C for 30 min, protected from light. Labeled Raw 264.7 cells were seeded onto the pre-treated bEnd.3 cells without any coating at a density of 5 × 10^5^ cells/mL and incubated for 1 h at 37 °C in the dark. Nonadherent Raw 264.7 cells were gently washed with PBS three times, and adherent macrophages were examined using an Olympus IX73 epifluorescence microscope. 

### 2.12. Data Expression and Analysis

All results were expressed as the mean ± SEM or mean ± SD, as appropriate. Data were analyzed using an one-way ANOVA followed by Sidák’s multiple comparisons test. Quantification of target proteins in Western blot analysis and of fluorescent macrophages in cell adhesion assays was carried out using Image J software, version 1.53. Flow cytometry results were analyzed with FlowJo Single Cell Analysis Software v10 (BD, Ashland, OR, USA).

## 3. Results

### 3.1. SNP Increases NO Production in a Dose-Dependent Manner without Altering Cell Viability

Preliminary experiments were carried out on bEnd.3 cells to determine the concentration of SNP. As seen in Figure 1a, none of the tested concentrations (1–1000 µM) were cytotoxic on bEnd.3 cells, as determined by an MTT assay. SNP increased NO production in a dose-dependent manner (Figure 1b). We chose to use 100 μM as the working SNP dose, a dose in the lower range of the tested concentrations that significantly increased NO levels. 

### 3.2. SNP Treatment Reduced LPS-Induced Alterations in Cellular Shape and/or Proliferation without Affecting Cell Viability

As shown in Figure 2a, LPS did not affect the viability of bEnd.3 cells after 24 h of treatment. Changes in cellular morphology and proliferation were monitored with real-time measurements of the cellular impedance of non-confluent bEnd.3 cells. Exposure of bEnd.3 cells to LPS significantly increased their nCI (Figure 2b). This increase was significantly counteracted by SNP treatment.

### 3.3. SNP Reduced ROS Production in LPS-Challenged bEnd.3 Cells and Increased NO Production

Intracellular ROS production was measured using flow cytometry with CM-H2DCFDA. As seen in Figure 3, LPS increased ROS levels in a significant manner when compared to control cells. SNP treatment significantly reduced LPS-induced ROS generation. Also, 24 h treatment with SNP significantly increased NO levels in bEnd.3 cells in the presence or absence of LPS, which has no effect on its own (Figure 4).

### 3.4. LPS-Triggered IL-6 Expression through the Activation of JAK2/STAT3 Pathway and SNP Attenuated LPS-Induced IL-6 mRNA Expression

We tested the effect of SNP on the production of IL-6, a pro-inflammatory marker, on LPS-challenged bEnd.3 cells. Results obtained from RT-qPCR showed that LPS treatment significantly induced a 10-fold increase in IL-6 mRNA expression after 24 h (Figure 5a). This increase was significantly counteracted by SNP treatment. 

To elucidate the signaling pathway that SNP triggers to exert this anti-inflammatory effect, a Western blot analysis of several proteins related to various pro-inflammatory pathways was performed. As showcased in Figure 5b, iNOS protein levels were not affected by treatment with LPS or SNP. On the other hand, a 24 h exposure to LPS induces a significant increase in STAT3 phosphorylation, an effect that SNP tends to decrease, although not significantly (Figure 5c).

### 3.5. SNP Treatment Fails to Reduce LPS-Induced Barrier Disruption

The potential protective effects that SNP has on LPS-challenged barrier integrity were assessed. As shown in Figure 6a, 24 h exposure to LPS significantly reduced bEnd.3 monolayer TEER. Additionally, treatment with SNP, whether alone or in the presence of LPS, resulted in a significant reduction in TEER.

The effect of LPS and SNP on the levels of tight-junction proteins such as claudin-5 and ZO-1 was also evaluated. After a 24 h treatment with LPS, claudin-5 protein levels were significantly decreased, an effect that was not reversed by SNP (Figure 6b). This decrease was verified by claudin-5 immunostaining (Figure 7).

The protein levels of ZO-1 increased after 24 h exposure to LPS. This increase was significantly reversed by SNP treatment (Figure 6c). Under physiological conditions, ZO-1 is situated at the cell membrane. After the LPS insult, no significant changes were observed in ZO-1 localization. However, some alterations in the cellular morphology were observed. When stimulated with LPS, bEnd.3 cells showed a shift from an elongated shape to a rounded shape (Figure 7).

Even though SNP treatment did not increase claudin-5 levels or alter ZO-1 localization, bEnd.3 cells regained their typical elongated morphology. Some changes in the actin cytoskeleton were also observed after SNP treatment. F-actin is evenly distributed across the cytoplasm under control conditions. However, when exposed to SNP, F-actin cytoskeletal undergoes rearrangement, and higher concentrations of actin filaments are observed in the cellular periphery (Figure 7).

### 3.6. SNP Treatment Does Not Prevent Inflammation-Induced Cellular Adhesion

A 24 h exposure to LPS significantly increased the adhesion of Raw 264.7 cells to the bEnd.3 cell monolayer, as displayed in Figure 8a,b. SNP treatment failed to prevent macrophage adhesion, as it tends to increase cellular adhesion by itself. 

We also tested the effect of SNP on the expression of two adhesion molecules, VCAM-1 and ICAM-1, on LPS-challenged bEnd.3 cells. Results from RT-qPCR showed a significant increase in VCAM-1 and ICAM-1 expression after the LPS insult. Under these conditions, SNP was able to revert LPS-induced VCAM-1 mRNA expression (Figure 8c), while it failed to do the same when evaluating ICAM-1 mRNA levels (Figure 8d).

## 4. Discussion

In various neurodegenerative pathologies, inflammation contributes to BBB disruption, making it a potential target for therapeutic interventions. NO is a crucial signaling regulator of vascular function; however, it cannot be strictly categorized as either an anti-inflammatory or pro-inflammatory molecule. The level of iNOS expression and NO formation determine whether it exerts beneficial or deleterious effects [19]. Furthermore, NO derived from eNOS regulates microvascular permeability in postischemic tissues and during inflammation [20]. 

Limited research has been conducted on the impact of NO donors on BBB permeability, and their role in inflammation and BBB disruption remains a subject of debate (see introduction). Thus, the main objective of this study was to explore the effects of SNP on LPS-induced inflammation and its potential role in protecting the BBB. Cerebral microvascular endothelial cells are the main luminal components of the BBB. We used the bEnd.3 endothelial cell line due to its recognition as a suitable model for the endothelial component of BBB since these cells appropriately express TJ proteins and form functional barriers to sucrose, which serves as a marker for permeability [21,22].

Primarily, our aim was to establish the effects of LPS on inflammatory markers and BBB permeability. LPS has been used as an inflammatory inducer in various cellular models, including endothelial cells. However, the results found in the literature are often controversial, and previous findings from other research groups described opposite effects of LPS on bEnd.3 cell viability. Although there are reports indicating that a 24 h treatment with LPS (100 ng/mL) could increase apoptosis of bEnd.3 cells [23], it has also been shown that exposure to LPS (1 µg/mL) had no significant impact on cell viability [24]. For our experiments, a dosage of 10 µg/mL LPS was chosen due to its lack of influence on cell viability (Figure 2a), yet it resulted in an elevation in the nCI (Figure 2b), suggesting a potential association with endothelial damage. A comparison of LPS doses (250 ng/mL–10 µg/mL) on the effects of nCI in bEnd.3 cells is available in the Appendix A.

LPS (10 µg/mL) treatment for 24 h led to an increase in ROS production (Figure 3), according to previous results showing the inhibition of the antioxidant enzymes glutathione peroxidase, catalase, and SOD in bEnd.3 cells after the LPS treatment. Furthermore, LPS decreases the reduced/oxidized glutathione (GSH/GSSG) ratio, indicating increased oxidative stress [23].

Oxidative stress and inflammatory responses are often linked. In our experiments, LPS significantly increased IL-6 gene expression and pSTAT3 expression, this last effect suggesting the activation of the JAK2/STAT3 pathway (Figure 5a,c). 

Previous studies suggest that IL-6 induction depends on STAT-3 activation in endothelial cells [25], while other authors have reported that IL-6 release activates STAT-3 via phosphorylation of Tyr705 through the JAK signaling pathway [26]. Furthermore, LPS stimulation in mouse BMECs led to the production of inflammatory factors such as IL-1b, IL-18, IL-6, and TNF-a [27].

Various studies point to the NO pathway as a potential signaling cascade that could mediate BBB disruption, and the inducible isoform of NOS (iNOS) is overexpressed during inflammation in different cell models, including bEnd.3 [28,29]. Although LPS increases NO formation by upregulating iNOS expression in microglia or macrophages, the results in endothelial cells are different. LPS (0.01–100 µg/mL) reduced NO production in the murine aortic endothelial cell line END-D [30]. Additionally, previous results showed no modification in NO production in bEnd.3 cells after LPS treatment (1 µg/mL) [24]. Surprisingly, other authors have described an increase in iNOS expression in bEnd.3 cells under the same conditions [31]. Our results using LPS (10 µg/mL) are consistent with studies reporting no change in NO production (Figure 4), and this is supported by the absence of modification in iNOS expression (Figure 5b). The detected NO is likely generated by eNOS, which is constitutively expressed in these cells.

The mechanisms by which LPS affects BBB function are not well understood. Several studies have demonstrated that an excessive increase in ROS reduces the expression of TJ proteins in endothelial cells. Consequently, this increase in ROS results in elevated BBB permeability, which is associated with the development of nervous system-related diseases [32,33]. For instance, in brain microvascular endothelial cells (BMECs), exposure to LPS led to an abnormal increase in ROS levels, resulting in a significant decrease in occludin and claudin-5, as well as TEER. These changes signify the impairment of BBB integrity, as reported previously [34]. Moreover, it has been observed that ROS induced by LPS can activate other molecular pathways, such as the RHOA/phosphoinositide 3-kinase (PI3K)/protein kinase B (AKT) pathway, which further reduces the expression of occludin and claudin-5, leading to a decrease in TEER [35]. Additionally, a maximal IL-6 response promotes the loss of junctional localization of VE-cadherin and ZO-1, and JAK-mediated STAT-3 phosphorylation at Y705 steadily increases permeability in endothelial cells [36].

It should be noted that in bEnd.3 cells, the effects of LPS treatment (1 µg/mL) appear to downregulate the expression of ZO-1, with controversial effects on occludin and no effect on claudin-5 [29,37]. Our findings using LPS (10 µg/mL) align with those observed in BMECs regarding claudin-5, demonstrating a significant decrease (Figure 6b) but an increase in ZO-1 expression (Figure 6c) accompanied by a change in the morphology of cells (Figure 7). These effects could explain the LPS-induced decrease in TEER (Figure 6a). Also, these changes in bEnd.3 morphology after LPS insult could be responsible for the increased nCI (Figure 2b).

Next, the effects of SNP on these alterations in inflammation mediators and BBB disruption induced by LPS were studied. As previously mentioned, NO exhibits both pro- and anti-inflammatory properties, depending on its localization and concentration. It has been described that localized suppression of NO production exacerbates inflammation, while, in contrast, local NO supplementation reduces leukocyte infiltration, vascular permeability, and the activity of enzymes related to oxidative stress [38]. Preliminary experiments were conducted on bEnd.3 cells to determine the dose of SNP for supplementation. We verified that a concentration of 100 µM significantly increases NO levels without inducing toxicity, as demonstrated by the MTT test (Figure 1). We also observed that SNP (100 µM) reduces ROS production after LPS treatment (Figure 3) as well as IL-6 production (Figure 5a), suggesting that SNP was able to mitigate this inflammatory status.

Different studies have shown that the reduced production of ROS by NO may be mediated by antioxidant enzymes whose expression is inhibited by LPS [38,39]. Special attention should be given to the SNP concentration since elevated concentrations (800 µM) have been used to induce oxidative damage in human umbilical vein endothelial cells (HUVEC) [40]. A comparison of SNP doses (1 µM–1000 µM) on ROS production in bEnd.3 cells is available in the Appendix A.

On the other hand, it has been reported in endothelial cells that NO donors, such as DETA-NO, reduce NF-κB activation and TNF-α-induced expression through AMPK activation [41], which may be related to the reduction in IL-6 expression observed in our experiments. While the decrease in IL-6 levels following pretreatment with SNP could potentially explain the reduced activation of STAT-3, results in other cell models, such as microglia, have indicated the regulation of STAT3 by NO-based post-translational modifications [42]. According to these results, we have not observed changes in the expression of p-STAT3 after a 6 h exposure to LPS in the presence or absence of SNP (Appendix A).

Regarding the effects of NO donors on BBB disruption, Wong et al. [43] showed, using human brain microvessel endothelial cells, that NO donors, through cGMP production, may offer a promising therapy for CNS inflammatory disorders exhibiting BBB breakdown. In line with this, Kho et al., utilizing both in vitro and in vivo models, have recently shown that treatment with the NO donor SNAP or sodium nitrate alleviates BBB breakdown [13]. On the contrary, a dose-related increase in the permeability of cultured rat brain endothelium has been reported following the application of SNAP [44]. Also, it has been found that the in vivo application of NO donors to cerebral microcirculation increases BBB permeability, and this could be related to peroxynitrite formation [16]. 

As previously mentioned, LPS-induced BBB disruption has been linked to an increase in ROS production or IL-6 expression. However, despite the fact that SNP (100 µM) counteracts these increments, it does not prevent damage to the barrier and even induces a significant decrease in TEER (Figure 6a). These changes in TEER values are consistent with what has been described by other authors in bEnd.3 cells using higher SNP concentrations (300 µM) [14]. On the other hand, it has been described that NO and subsequent cGMP may act through Ca^2+^/calmodulin to increase the F-actin contents, thus resulting in actin reorganization [45], as can be observed in Figure 7.

NO regulates the expression of adhesion proteins on the endothelium, and the transcriptional regulation of these proteins is controlled by NF-κB, whose activation leads to the de novo synthesis of high levels of messenger RNA for E-selectin, P-selectin, ICAM-1, and VCAM-1. Low NO concentrations, slightly above basal levels, increase NF-κB activity, whereas higher concentrations inhibit it [46]. In HUVEC, it has been shown that low concentrations of the NO donor nitroglycerin (50–100 µM) increase the expression of the adhesion proteins ICAM-1, VCAM-1, and E-Selectin, while higher concentrations (250–500 µM) reduce it [47]. 

In our experiments, mRNA expression of VCAM-1 and ICAM-1 was not modified by SNP, and it was significantly increased by LPS. This LPS-induced increase in VCAM-1, but not in ICAM, was counteracted by SNP (Figure 8c,d). These findings could explain why SNP did not significantly alter the adhesion of Raw 264.7 cells to bEnd.3 cells induced by LPS (Figure 8a,b).

## 5. Conclusions

Inflammation is associated with BBB disruption in neurodegenerative diseases, making it a potential therapeutic target. NO is a crucial regulator of vascular function, but its effects are context-dependent, and the level of NO formation determines whether it has beneficial or deleterious effects. In our experimental model, LPS treatment led to an increase in ROS production and an inflammatory response, including activation of the JAK2/STAT3 pathway and the expression of pro-inflammatory markers like IL-6. SNP, an NO donor, reduces ROS production and IL-6 levels, potentially mitigating the inflammatory response (Figure 9). However, it does not prevent BBB disruption induced by LPS, leading to decreased TEER and altered expression of TJ proteins like claudin-5 and ZO-1. Also, the concentration of SNP used in this study did not significantly alter the adhesion of Raw 264.7 cells to bEnd.3 cells but showed a trend towards an increase. Our findings suggest that while SNP demonstrates some anti-inflammatory effects, these effects may not completely translate into direct protection of the BBB in our experimental conditions. Therefore, the potential use of SNP in neurodegenerative diseases, while not entirely ruled out, should undergo further in-depth investigation.

## Figures and Tables

**Figure 1 cells-13-00843-f001:**
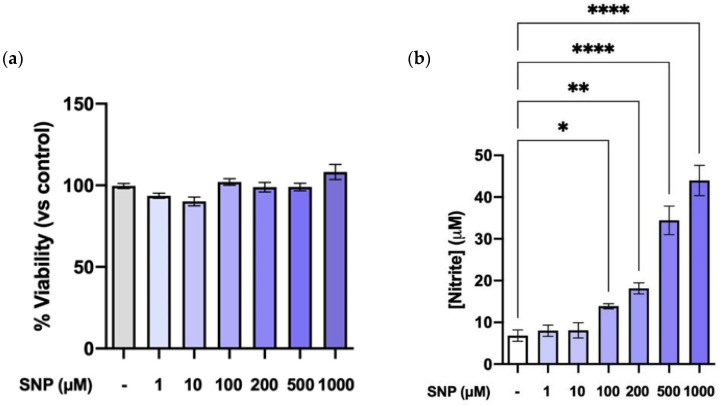
Effect of increasing concentrations of SNP in cell viability and NO production. (**a**) Cell viability of bEnd.3 cells exposed to increasing SNP concentrations determined by the MTT assay. (**b**) NO production by bEnd.3 cells after treatment with increasing concentrations of SNP measured using the Griess assay. Results are given as mean ± SEM; n = 6 technical replicates. All the experiments were repeated at least four times. * *p* < 0.05, ** *p* < 0.01, and **** *p* < 0.0001, as indicated in the graph.

**Figure 2 cells-13-00843-f002:**
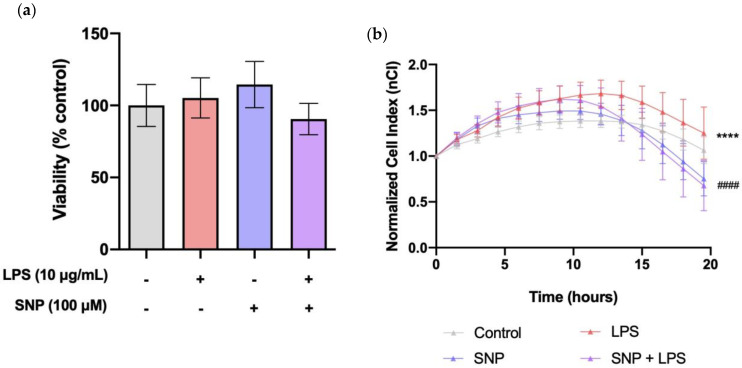
SNP mitigated LPS-induced alterations in cellular shape and/or proliferation without affecting cell viability. (**a**) Cell viability of LPS-challenged bEnd.3 cells with and without SNP co-treatment was determined by the MTT assay. (**b**) Impedance-based real-time detection of the effect of SNP on bEnd.3 morphology and/or proliferation after 24 h exposure to LPS. Results are given as mean ± SEM; n = 5 technical replicates. All the experiments were repeated at least four times. **** *p* < 0.0001 with respect to control, ^####^
*p* < 0.0001 with respect to LPS-treated cells.

**Figure 3 cells-13-00843-f003:**
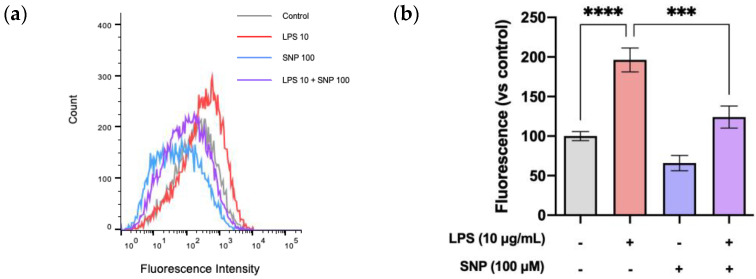
SNP reduced ROS production in LPS-challenged bEnd.3 cells. (**a**) Representative histograms of ROS production by bEnd.3 cells. (**b**) Quantitative determination of ROS production in bEnd.3 cells after 24 h exposure to LPS with or without treatment with SNP. Results are given as mean ± SEM; n = 3 technical replicates. All the experiments were repeated at least four times. *** *p* < 0.001 and **** *p* < 0.0001, as indicated in the graph.

**Figure 4 cells-13-00843-f004:**
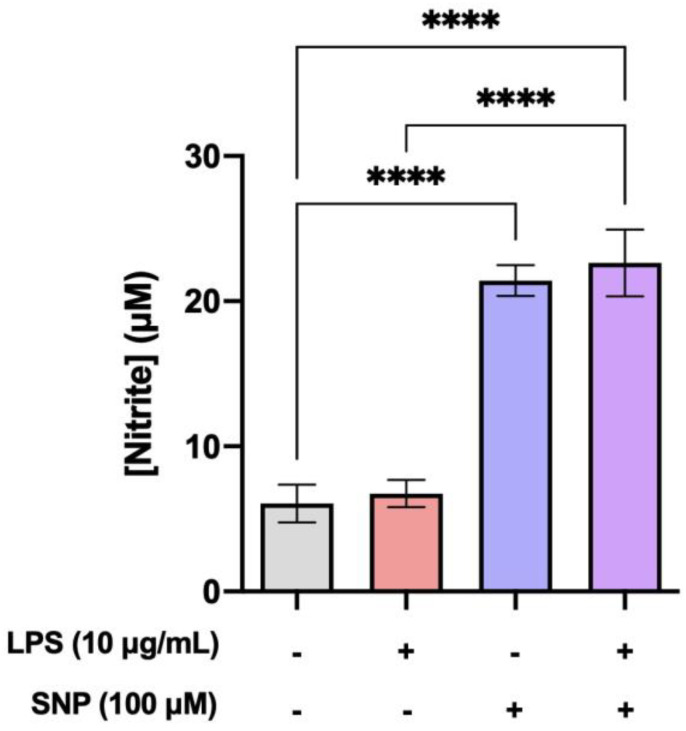
Modulation of NO production by LPS and NPS. Effect of SNP on NO production after 24 h exposure to LPS. Results are given as mean ± SEM; n = 5 technical replicates. All the experiments were repeated at least four times. **** *p* < 0.0001, as indicated in the graph.

**Figure 5 cells-13-00843-f005:**
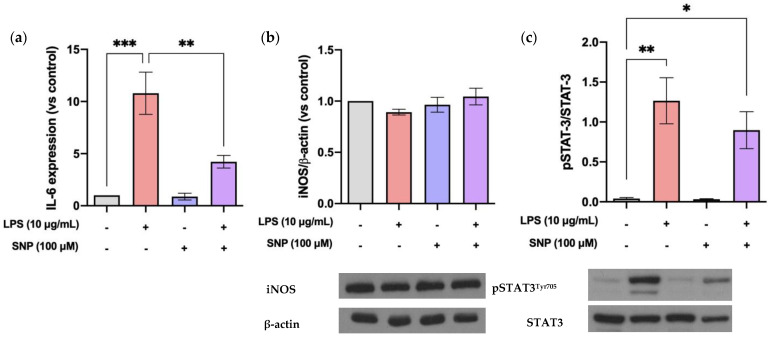
LPS triggers IL-6 expression through the activation of the JAK2/STAT3 pathway, and SNP counteracts this effect. (**a**) Effect of SNP on IL-6 mRNA expression after 24 h exposure to LPS. (**b**) iNOS protein levels were examined by Western blot analysis after 24 h exposure to LPS in the presence or absence of SNP. (**c**) STAT3 phosphorylation after exposure to LPS in the presence or absence of SNP for 24 h. Results are given as mean ± SD; n = 3 technical replicates. All the experiments were repeated at least four times. * *p* < 0.05, ** *p* < 0.01, and *** *p* < 0.001 vs. control or vs. LPS, as indicated in the graph.

**Figure 6 cells-13-00843-f006:**
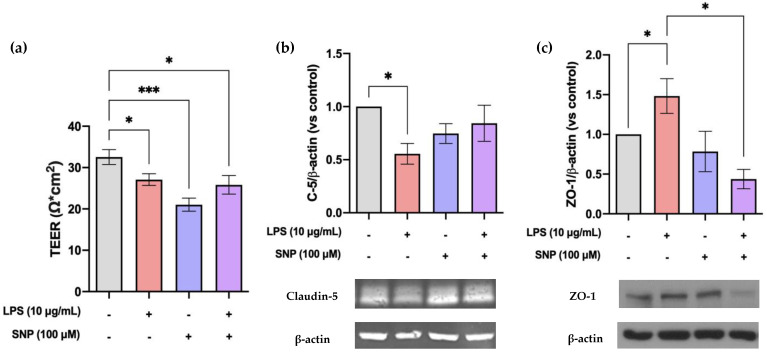
SNP treatment fails to reduce LPS-induced barrier disruption. Effects of SNP on bEnd.3 (**a**) TEER, (**b**) claudin-5 protein levels, and (**c**) ZO-1 protein levels after 24 h exposure to LPS. Results are given as mean ± SD; (**a**) n = 3 technical replicates. All the experiments were repeated at least four times. * *p* < 0.05 and *** *p* < 0.001 vs. control vs. LPS, as indicated in the graph.

**Figure 7 cells-13-00843-f007:**
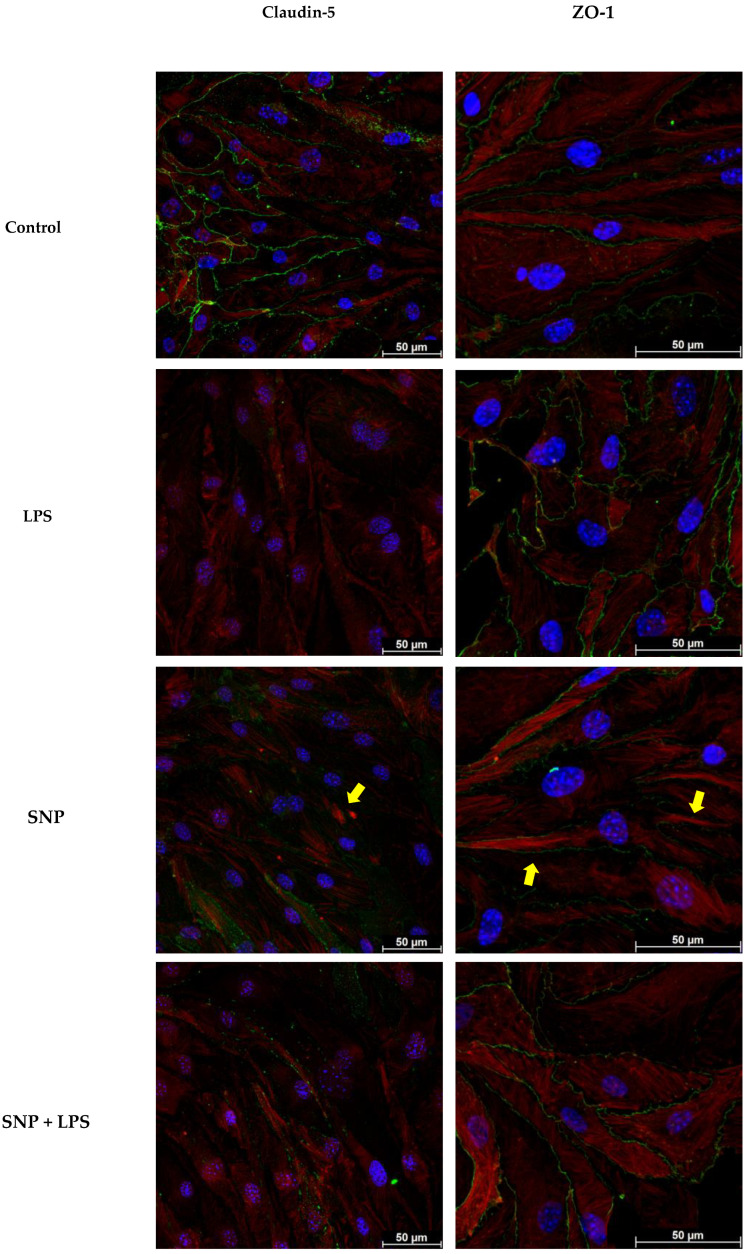
Representative fluorescence microscopy images of claudin-5 and ZO-1 immunostaining after 24 h exposure to LPS and/or SNP. Yellow arrows indicate zones of higher actin filament concentrations.

**Figure 8 cells-13-00843-f008:**
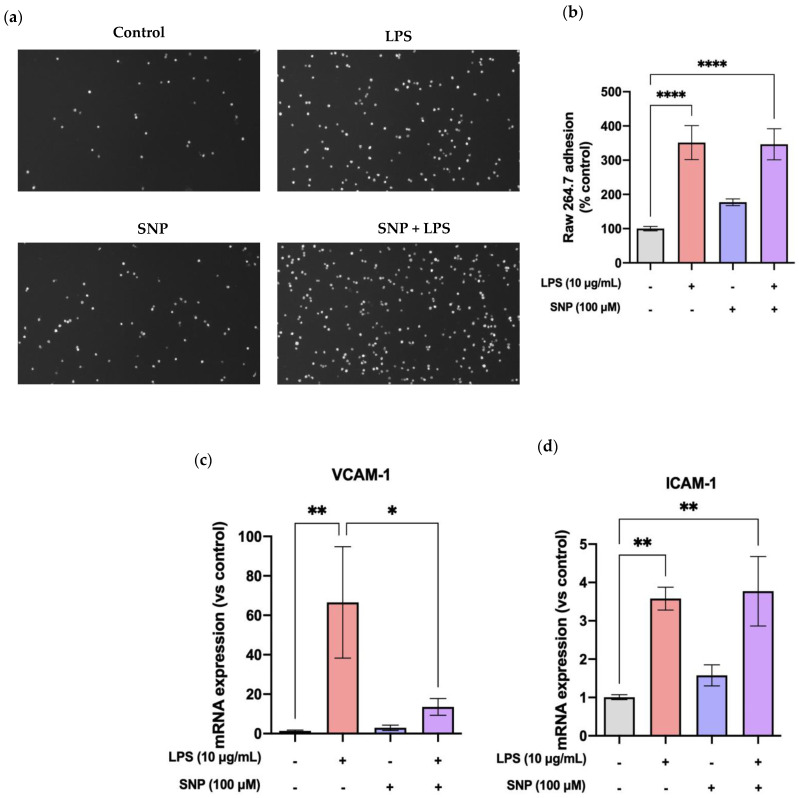
LPS treatment increases cellular adhesion. (**a**) Calcein AM-labeled Raw 264.7 macrophages adhered to LPS-challenged bEnd.3 cells in the presence or absence of SNP treatment. (**b**) Effect of SNP on the percentage of Raw 264.7 adhesion to LPS-challenged bEnd.3 monolayers. (**c**) VCAM-1 and (**d**) ICAM-1 mRNA expression in the presence or absence of SNP after exposure to LPS for 24 h. Results are given as mean ± SEM; n = 3 technical replicates. All experiments were repeated. * *p* < 0.05, ** *p* < 0.01, and **** *p* < 0.0001.

**Figure 9 cells-13-00843-f009:**
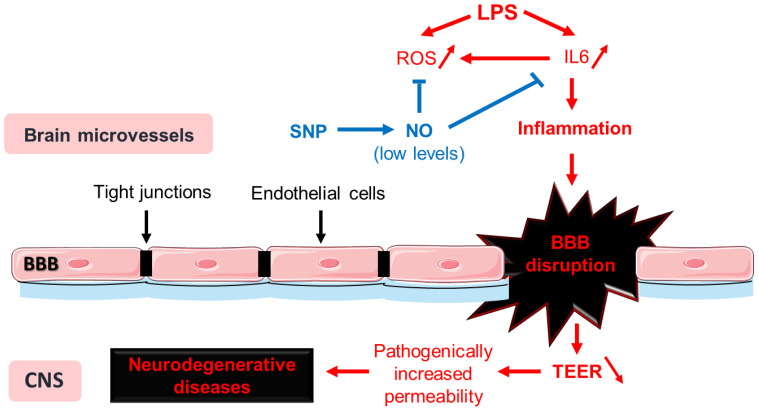
In our experimental model, LPS-induced inflammation triggers ROS production and pro-inflammatory marker expression, an effect that is mitigated by SNP. Despite its anti-inflammatory effects, SNP may not directly protect the blood–brain barrier under our experimental conditions, deserving further investigation for its potential use in neurodegenerative diseases.

## Data Availability

The raw data supporting the conclusions of this article will be made available by the authros on request.

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
