# Peer review of "Effects of Sodium Nitroprusside on Lipopolysaccharide-Induced Inflammation and Disruption of Blood–Brain Barrier"

_cells, 2024, doi:10.3390/cells13100843_

Round 1
Reviewer 1 Report (Previous Reviewer 3)
Comments and Suggestions for Authors
I appreciate the modifing version of the paper and I think that the supplementary material, supplied by the authors according with the reviewers, has improved the quality of this research work.
Reviewer 2 Report (Previous Reviewer 1)
Comments and Suggestions for Authors
The authors have addressed the questions raised in the first round of review
This manuscript is a resubmission of an earlier submission. The following is a list of the peer review reports and author responses from that submission.
Round 1
Reviewer 1 Report
Comments and Suggestions for Authors
The article provides a comprehensive understanding of sodium nitroprusside’s effects in context of inflammation and BBB disruption. It covers all the necessary questions and lays out the findings logically. It does a good job overall and provides valuable insights.
My specific comments and suggestions are outlined below:
1. The authors do provide a rational for using the LPS dose of 10 µg/ml. Is this the only dose of LPS that has been tested by the group? Is there a LPS dose comparison that has been done? eg. 10 vs 100 µg/ml effect on nCI? Or LPS 10 vs 100 µg/ml and SNP treatment effects on ROS / NO production or tight junction. I believe that could be helpful to solidify the trends.
2. “We chose to use 100 μM as the working SNP dose, a dose on the lower range of the tested concentrations that increased NO levels without affecting bEnd.3 cell viability.”
Figure 1A shows viability is unaffected regardless of dose of SNP. If authors could elaborate on why choice of lowest dose that increased NO levels is important, that would be helpful to the readers. have the authors tried different high vs low doses of SNP to see their impact?
Reviewer 2 Report
Comments and Suggestions for Authors
The authors of the article: “Effects of sodium nitroprusside on LPS-induced inflammation 2 and disruption of blood-brain barrier”, included a comprehensive analysis of the issue at the in vitro level. Even though the hypothesis was negatively verified, the work is valuable. It will help avoid repeating similar studies by other teams dealing with the functioning of the blood-brain barrier. Moreover, the work seems well-thought-out and well-organized, so I have no major objections to the content and presentation of the results. I only have a few minor comments on the text:
1) Section: 2.2. Cell culture and treatment
Please specify the basis for selecting the SNP dose
2) Section: 2.9. Western Blot analysis and 2.10. Immunocytochemistry
Please provide catalog numbers and names of manufacturers of the antibodies used.
Reviewer 3 Report
Comments and Suggestions for Authors
Introduction: This section appears clear and well written. I suggest the authors to insert a summary scheme that can summarize all the reported effects of the molecular mediators in the study.
Materials and methods:
Include a wider toxicity study on lps in endothelial cells used, so that the choice of authors between different doses and times is justified and supported by data
Cell adhesion assay: describe the preparation of the plate and the matrices used for the adhesion study, was a particular coating applied to avoid the non-specific adhesion of macrophages?
Line 238: If no dose induces cytotoxicity say that you choose the dose that does not interfere with viability is superfluous.
MTT assay: Experimental times should always be reported
Figure 2: The data in this figure are not very clear and solid:"SNP mitigated LPS-induced alterations in cellular shape and/or Proliferation without affecting cell viability", but the combination of snp and lps determines a decrease in vitality. in any case the modulation of vitality is insignificant and very mild. Moreover in figure two the curve of the LPS has a course strangely comparable to the control, to the contrary snp and the combination seems to undergo a rapid decline, to discuss better this effect or to re-evaluate these data.
I also suggest that authors demonstrate the induction of inflammation mediated by the chosen dose of LPS, using a western blot for COX2 or PGE2, in order to support the main purpose of the manuscript
Fig5: Better describe the statistical analysis, as it appears first vs the control and then vs LPS: The terms of comparison on which the calculation of p value is based must always be explained in ALL captions
Is there any evidence that STAT3 was obtained more quickly? The phosphorylation of STAT3 is a rapid event and NO is a gas mediator that should perform its function even in a short time, if the authors have given a short time I suggest to consider it possible to insert them to better deepen the mechanism studied.
Fig 6a: Describe the statistical analysis, is vs ctr or vs LPS?
Increase the quality of WB images or change with another set of blots
Immunofluorescence: Immunofluorescence: there are no substantial differences in the chosen images, try to choose images where the difference in zo-1 or actin is more evident or indicate with arrows the points that support the thesis discussed by the authors. It would also be possible to support this data with a wb for the same markers.
Line 514-515: "However, when exposed to SNP, higher concentrations of actin filaments are observed in the cellular periphery" Please, add a short period in which you define the meaning of this different localization of actin and the importance of this difference in the analyzed context.
Discussion: The discussion is well written but needs to be revised on the basis of the data I suggested to review.
